# Organizational and Supply Chain Impacts of 3D Printers Implementation in the Medical Sector

**DOI:** 10.3390/ijerph19127057

**Published:** 2022-06-09

**Authors:** Fabio Musso, Federica Murmura, Laura Bravi

**Affiliations:** Department of Economics, Society, Politics, Carlo Bo University of Urbino, Via Saffi 42, 61029 Urbino, Italy; fabio.musso@uniurb.it (F.M.); federica.murmura@uniurb.it (F.M.)

**Keywords:** medical 3D printing, Industry 4.0, additive manufacturing, healthcare management

## Abstract

3D printing application extends to various sectors, such as aerospace, construction, art, domestic, up to healthcare. It is in this domain that its adoption could offer technological solutions aimed at improving the individual life and guaranteeing organizational effectiveness. The aim of this study is to understand the way in which the adoption of medical 3D printers has introduced economic-business changes at the supply chain, organizational and environmental level within business processes considering the point of view of 3D printer manufacturers. A multiple case study has been developed, through the administration of a semi-structured interview to 7 Italian companies that design, manufacture and sell 3D printers offering additive technological solutions to the medical sector. The results show how companies believe that the organizational impact related to the adoption of this technology is quite significant, highlighting how it leads to the definition of a new organizational culture. Secondly, it emerges that the adoption of 3D printers within the medical sector also leads to a change in procedures and production activities. Finally, it also emerges that the impact at the supply chain level particularly affects the reduction in the number of players in the supply chain and product time to market.

## 1. Introduction

Responding to market global needs has led organizations to move their way of doing business mainly due to the changing scenario of recent years. In addition to the changing of consumer demand, even more personalized and looking for specific products, also the way in which companies relate to the same has changed, being more attentive, more informed and more demanding. The relationship that has been created in recent years between business and consumer is of one-to-one type and leads to the creation of a single product for each individual, often involving the person concerned in the production process [1]. According to scholars, recently the single tangible product has become part of an ecosystem that involves numerous organizations and a growing number of interconnected products and services, and all together become indispensable for the final offer and for the value creation [2]. The actual economic context has radically changed the meaning of technology, which from a simple production factor has become a critical competitive factor capable of changing business strategies and revolutionizing entire sectors and production processes [3].

Therefore, the question is whether these changes may lead to the possibility of considering a total or partial reconfiguration of manufacturing activities, thinking of a production mode oriented to an approach that is no longer subtractive, but of an additive type [4,5].

Additive Manufacturing (AM), better known as 3D printing, is a great innovation in every area of production, and as it is easy to guess its introduction has led to profound changes and profound impacts at the organizational level and on business processes. 3D printing is a production technique that is placed within a much broader context, namely that of enabling technologies defined by Industry 4.0: in this scenario, which well-known scholars such as Schwab and Rifkin define as revolutionary [6,7,8], it is witnessing the adoption of technologies capable of impacting the entire business system, giving rise to a series of economic implications. Its application extends to various sectors, such as aerospace, construction, art, domestic and many others, up to healthcare; it is precisely here that the adoption of this technology can mark the beginning of a new era by offering technological solutions aimed at improving the life of the individual and at the same time guaranteeing organizational effectiveness from the point of view of costs to be incurred and time employed in the realization [9].

The orthopedic sector is one of the areas in which the use of these technologies finds greater application, an example in Italy is provided by Dr. Guido Grappiolo, who, in collaboration with his team, has implanted a hip of metal material created and printed in 3D; from that moment on, another 600 prostheses were implanted, benefiting individuals (https://www.humanitas.it/ accessed on 16 February 2022). Clearly, according to the use made, different materials can be used for the production: if it is necessary to produce components to be implanted inside the human body, the material should be compatible as well as responding to the strict requirements set by international standards, otherwise it is not necessary [10]. The 3D printing of inorganic objects and not intended for implantation in humans, on the other hand, has certainly proved useful for studying and better understanding the anatomical complexity of some pathologies that require surgery. The use of such a 3D printed model allows cardiac surgeons to manipulate the organ before performing the operation and to familiarize themselves with the specific anatomy for a particular patient. This confers a strong advantage in preparing for the surgical procedure [11].

Furthermore, the current health emergency deriving from the COVID-19 pandemic has accentuated the relevance of the use of AM in the medical field. From the very outset of the COVID-19 pandemic, healthcare and personal protective equipment (PPE) suppliers began to struggle to meet the acute demands of specific items such as face masks, face shields, test kits, ventilators, etc. [12]. The existing production capacities could not cope with the unexpected and acute demands. Moreover, global supply chains were disrupted as a result of reduced employees and lockdown in many areas of the world, making the situation even more critical [13]. In this scenario, AM had a relevant role in the fight against COVID-19 by producing components for medical equipment such as ventilators, nasopharyngeal swabs and PPE such as face masks and face shields [14].

Starting from these assumptions, this study has the aim to analyze from the point of view of 3D printer manufacturers, what are their perceptions regarding the organizational impacts generated by 3D medical devices.

More specifically, the aim is to analyze how 3D printing impacts on the business organization of the purchase companies and on the entire supply chain, highlighting current and future opportunities, and critical issues deriving from the adoption of this technology.

The research questions that explicitly emerge are the following:


*RQ1: What are the most significant impacts generated by the adoption of 3D printers in the medical sector at the business organizational and supply chain level?*



*RQ 2: What current and future opportunities and what critical issues are generated by the use of this technology in the medical sector?*


The originality of the study lies in the sector considered, that is the medical one. There are no relevant studies in the literature panorama that consider the organizational and supply chain changes that can derive from the use of 3D printing in the medical sector. This study can be seen as a forerunner for further research developments in this area.

The rest of the paper is structured as follows: Section 2 analyzes the literature on the impact of 3D printer at the business organizational and supply chain level, highlighting the criticalities and weaknesses that emerged from past studies; Section 3 defines the methodology used, Section 4 presents the main results obtained, while Section 5 discusses the results and draw the main conclusions.

## 2. Literature Review

### 2.1. Impact of 3D Printing at the Supply Chain Level

The supply chain can be defined as “a set of entities (e.g., organizations or individuals) directly involved in the supply and distribution flows of goods, services, finances, and information from a source to a destination (customer)” [15]. This process begins with the raw materials, continues with the creation of the product, which passes through the hands of the intermediaries (wholesalers and retailers), and ends with the supply of the final product to the customer. The introduction of additive manufacturing leads companies to adopt a new production paradigm and therefore, it is necessary to reformulate the supply chain capable of managing manufacturing activities [16]. With the introduction of 3D printing into the production process, there are a series of changes in warehouse management, in the number of actors involved in the chain and in activities related to production volumes: in addition to determining the transition from a production of mass scale (based on standardized goods) to additive manufacturing, (based on the creation of customized products at lower costs), 3D printing offers a way to simplify the production chain by determining a less complex supply chain and significantly reducing it [17]. For example, through additive manufacturing assembly is less necessary as there are products that can be printed in one piece; this implies less coordination of suppliers and a consequent reduction in costs and time. Otherwise, with traditional production methods it is necessary to produce each component separately and then assemble them together at a later time [18]. The biggest change introduced by the 3D printer concerns the displacement of production in space, in fact, after having designed the model of a product, through the transfer of the file via a simple internet connection, it is possible to produce the object in question anywhere in the world: in this way the number of intermediaries is reduced with a consequent saving in transport costs and also the time to market is reduced [16]. With reference to the medical field, this is precisely what happened during the first months of the COVID-19 pandemic: the technology allowed the transfer to thousands of computers of the 3D model of the design of a face mask and a component of the cooling fan. Rather than waiting for medical components to first be mass-produced abroad and then shipped to hospitals, 3D printing made it possible to manufacture them nearby and on demand [19]. Additive manufacturing applied to the medical sector has also led to a reduction in the relocation of the supply chain since health facilities and hospitals, instead of sourcing from foreign companies, can produce medical equipment and components directly within the hospital itself [20], giving a relevant improvement in aligning and streamline the entire health care supply chain from providers to purchasers [21].

In addition to the creation of highly customized and on-demand products, additive manufacturing also leads to profound changes in the management of internal logistics and in the production of spare parts. This technology leads to a series of advantages in relation to the reduction of waste as it reduces the need to hold large quantities of products or spare parts in the warehouse; all this because 3D printing allows to obtain unique products in small series, making it possible to print the products at the right time and only on specific request. According to this logic, instead of keeping spare components in stock, companies could hold files that store the different geometries of the components to be created, creating a sort of “virtual warehouse”, significantly reducing logistics costs [18].

From an environmental point of view, although plastic is the most used material [22], companies using 3D printing are moving towards the use of more sustainable and environmentally friendly materials [23]. With additive technology, the amount of waste is also drastically reduced as it makes it possible to produce layer by layer, as opposed to the subtractive technique which produces a significant amount of waste.

The introduction of 3D printing in production processes marks the transition from a production oriented to “build to stock” to one to “build to order”: in the first case it is a question of basing production on market studies and demand estimates, in the second production takes place on request, it is produced only if the consumer places an order, in this way the stocks of products are reduced to a minimum, waste is reduced and the risks of overproduction are avoided [24].

### 2.2. 3D Printing Impacts at the Corporate Organization Level

When a 3D printer is introduced within the business organization, the main concern is raised by the fact that 3D printers, in all stages of production, do not require human intervention except in the initial configuration phase. In the event that the product produced is not a finished product, then it is possible that man can intervene in the assembly phase, but in the event that the printer reproduces a finished product it will perform its tasks in total autonomy. In any case, introducing a new machine into the processes does not mean that this necessarily replaces the work of man, but, with a change of perspective, it can be thought of as a complementary technology to man, which makes it easier and fast execution of complex activities.

The figure of the worker will always be present in the production cycle, in particular in the phases concerning the design, the implementation of 3D systems and in the management of the business, defining a form of flexible work organization and thus eliminating the old structural and technological constraints that often required a single choice between man and machine [25].

In addition to this form of flexible organization, in the medical field, the use of the 3D printer has led to a reorganization of roles as well as the birth of new “hybrid” figures within the hospital system, since the use of 3D printing is certainly a revolution that changes the way of doing medicine and within the production processes leads to innovation also of a cultural nature [26].

Within the health and hospital facilities, alongside surgical doctors and radiologists, there is the figure of the 3D specialist that is configured with the figure of the biomedical engineer who provides technical support in the design and production process and has led to major changes in the organizational system [27]. If you think, for example, of the creation of a 3D printed prosthesis, the collaboration of engineers and doctors is necessary since the design and application requires a series of integrated knowledge of physics, medicine and cell biology [28].

Considering the main criticalities of using Additive Manufacturing in the medical field, there is the difficulty in the identification of the most suitable 3D technology: for example, a supplier of healthcare products that decides to implement additive manufacturing in its processes must carefully select the most suitable technique in reference to its production chain and raw materials that allow an optimal reproduction of the object [29].

Another criticality is related to the training of personnel who must be highly qualified in order to be able to juggle highly sophisticated technologies: this aspect involves the incurring of costs for training and continuous refresher courses as stated by Weidert et al. [30].

From an ethical and social point of view, questions arise regarding the use of the press and the responsibilities of those involved in the event of malfunction of medical products. In the health sector, to date, there is still no specific legislation in the United States and Europe that regulates these products, which fall within the Research & Development activities; for a number of aspects, therefore, the 3D printing scenario is very fragmented especially in Europe, but it is expected that in the future a standardization of operations will be applied at an international level [31].

## 3. Materials and Methods

To understand the way in which the adoption of the 3D printing technology within production processes in the medical sector has introduced economic-business changes at the supply chain and organization level, it has been developed a multiple case study analysis of a descriptive type, as the research aims to delineate the general and particular features of a given phenomenon [32], through the realization of a semi-structured interview with 7 companies that design, produce and sell 3D printers offering additive technological solutions to the medical sector. It has been decided to adopt the methodology of a multiple case study [33], as it allows a clearer understanding and characterization of the investigated phenomenon, providing a strong and reliable evidence [34]. Moreover, the multiple case study allows a better generalization of results and also a direct comparison between the similarities and the differences emerged from the investigations carried out on the various phenomena considered.

The decision to analyze 7 companies is in line with Yin [33], according to which, the recommended number of units to be examined to analyze a good multiple case study is between 4 and 12.

The semi-structured interview, used to collect primary data, was set up as follows: during the first part of the interview, general questions were asked relating to turnover, number of employees, reference markets, etc., with the aim of outlining the profile of the company interviewed. In the second part, more specific questions were asked aimed at identifying the organizational impacts of 3D printers within the company at the supply chain and business organization level. During the interviews, the interviewees were asked to quantitatively evaluate, using a scale of importance from 1–5 (1 = not important at all; 5 = very important), some factors related to the impacts of the use of 3D printers both on supply chain and on the business organization of its customers, in order to have a quantitative response to the qualitative impressions discussed during the interview.

The interviews were carried out in November 2020, lasted about an hour and a half each and were based on an interview protocol; these were carried out with the Chief Executive Officers of the participating companies, or for smaller companies with the owner, looking for interviewing the position with the greatest technical and organizational skills of the company. Secondary data were gathered thanks to a review of the companies’ websites and their profiles on different social networks in order to comply with the triangulation principle and to validate the information gathered through the semi-structured interview.

### Survey Sample

The companies to which the semi-structured interview was administered produce and sell 3D printers offering solutions for the medical sector. Below is a brief description of each of the companies participating in the case study analysis.

*Business 1*: it is a leading company in the 3D printing sector, and designs, manufactures and sells printers entirely made in Italy. About 4 years ago, the company started a project which develops 3D printing solutions for the medical sector by offering skills and tools in order to create advanced medical devices. It not only works on prototypes, but also and above all on finished products ready for use as prostheses and orthoses as well as various kinds of aids such as braces or orthopedic braces. In addition to this it offers the possibility to purchase professional training courses and packages on 3D printing, to teach how to learn to relate to this technology at different levels (basic, intermediate and advanced course).

*Business 2*: it is also an Italian manufacturer and seller of 3D printers. It stands out for offering reliable multi-material printers (with double or triple extruder) equipped with a simple and intuitive interface. There are many companies that print with this business, such as Louis Vuitton, Decathlon and Airbus, and at the same time they are resellers in university graduate schools. In particular, their 3D printing solutions offered a 3-extruder technology that allowed them to offer tactile tools to students with visual impairments, allowing them to relate to this technology thanks to multi-colored and multi-material supports.

*Business 3*: it produces and sells 3D printers. The materials most used in their printing technologies are metal and resin, in particular this is used to produce a vast amount of biocompatible material with high levels of precision, favoring the production of devices in contact with biological tissues of the human body. Their technology also makes it possible to create dental models, again with high levels of precision through the use of biocompatible resin.

*Business 4*: it is a reality made up of a team of innovators and researchers (as well as engineers, technicians etc.) with a strong passion for 3D printing. The commitment of this company is to fully understand the development of technology as they believe it is destined to revolutionize the entire production system and not just the field of prototyping.

*Business 5*: specializes in the modeling of 3D objects and offers a wide selection of filaments for printing and special materials based on individual needs. It operates in various sectors including the medical one and also organizes workshops and courses to spread the technology and to create a solid training structure.

*Business 6*: is a company that operates in the automotive, aerospace and medical sectors, promoting treatments aimed at improving the quality of life. Their technologies also offer solutions in the dental sector, contributing to the growth of digital dentistry, and develop equipment to facilitate pre-operative surgical practice.

*Business 7*: it manufactures and sells 3D printers. In the medical sector, its printers allow the production of orthopedic prostheses and conceptual spinal implants in metallic material. The company operates all over the world even though its main office is in the United Kingdom, and it is there that it carries out most of its production and R&D activities, providing solutions in the medical, spectroscopy and manufacturing sectors.

The socio-demographic profile of respondent companies is presented in Table 1.

## 4. Results

Considering which 3D printing technologies have been implemented by the manufacturing companies interviewed, Table 2 shows that some companies make printers with multiple printing techniques and among these the most adopted by 5 out of 7 companies is that of Fused Deposition Modeling (FDM), followed by resin printing.

While considering businesses’ main customers, it can be seen that healthcare companies are the most frequent, followed by Universities. Business 3 works with dental prosthesis manufacturers, Business 2 also with other manufacturing businesses and Business 4 with privates too (Table 3).

From the analysis in Table 4 it emerges that Business 7; which is a micro-enterprise born in 2017, is firmly convinced that the use of 3D printers within the company organization produces very strong impacts on all the factors considered: in fact, it attributes the maximum score to all items as opposed to Business 3, which despite being a medium-sized company operating in international contexts, it attributes an intermediate value to all items, implying that, in its opinion, the level of importance of each impact is neither strong nor weak. The item “orientation to a new organizational culture” is the one that has linked all companies to some extent and which is given a greater weight of importance (with the exception of company 3). Overall, companies give high scores to the attributes under examination, confirming what they said in the interviews carried out: in addition to cultural change, they believe that the adoption of 3D printing leads to important changes in the management procedures of companies, changing routines and normal activities, and therefore at the same time this involves changes in the duties and tasks of the staff, leading to the need to introduce figures with new skills.

As concerning the impact on the supply chain of 3D medical printers’ buyers, the interviewees all agree that at least partially there have been significant changes. In detail as it can be seen from Table 5 additive manufacturing leads companies to adopt a new production paradigm and therefore, a reformulation of the supply chain is necessary. The reduction of actors in the supply chain, the reduction of time to market and inventory in the warehouse are the items to which companies have given greater importance during the interviews carried out and consequently are those with the highest scores in Table 5. On the basis of this, it is possible to confirm that with 3D printing within the production process, there is a series of changes related to the number of actors involved in the supply chain, that is, the number of intermediaries is reduced (with a consequent reduction in costs), and at the same time, time to market is reduced. The company that has attributed the highest scores to the items is Business1, that is the longest-running company founded in 1955: this leads to think that by operating for a long time, it has had the major opportunity to see market changes over the years and, therefore, also of the supply chain.

As regards current and future opportunities generated by the application of 3D printers in the medical field (see Table 6), each company attributes a weight of high importance to all the opportunities deriving from the adoption of 3D technologies, in the following order: the developments of highly customized materials aimed at improving the lives of individuals, followed by the possibility of new market outlets, the development of biocompatible materials, the birth of new specialized professionals and the development of new and increasingly sophisticated technologies. It is interesting to note that the development of the 3D printer within the medical system can lead to big revolutions that in future years will contribute to a significant improvement in people’s lives, also thanks to the development of biocompatible materials. This also leads to the deduction that the strong potential deriving from the adoption of these technologies is real, irrefutable and shared.

As for the criticalities and weaknesses of this technology, these are perceived by companies as very weak and not too much significant (see Table 7). In particular, ethical and social criticalities are almost completely non-existent, with the exception of Business 2 which attributes a higher score (4) and in the interview the CEO raises concerns about bioprinting, that is still a reality far from the one in which we live, but that in the future, it will have a markedly different impact. The most significant critical issues, on the other hand, are those relating to personnel training and the difficulty to adequately train them and in the non-regulation of the market. What can be said with certainty is that the opportunities generated by the adoption of 3D printing technologies in the medical sector are perceived as greater and more important than the critical issues they generate.

## 5. Conclusions

The purpose of this study was to examine the perception of organizational impacts by manufacturers and sellers of 3D printers in the medical sector, in particular by understanding the changes in the supply chain, and in business organization resulting from the adoption of this technology.

The fields of application of 3D printing are numerous, but the sector in which more research is carried out and which appears to be constantly growing is certainly the one related to healthcare, allowing additive manufacturing to become a leading production technique in the medical sector for a wide range of applications. Working in this context requires a vision oriented to a digital, farsighted and innovative perspective which, within the context in which it operates, leads to innovation also in the organizational culture of healthcare facilities [18]. Technologies adopted by companies in the medical field are different, and as shown in literature the most adopted technology is FDM.

As it concerns RQ1, the study shows that the general impact at the business organization level is quite significant, highlighting how the adoption of these technologies leads to a new organizational culture [37]. The use of 3D printing changes the way of doing medicine, leading to profound changes within production and distribution processes [26]. By transforming these concepts into concrete ideas, two fronts of innovation can be identified. The first refers to the possibility of creating customized solutions for bone and orthodontics prostheses and medical aids, increasing the number of possible applications and modifying production techniques. In this way, there is a tendency to overcome the traditional industrial setting, which starts from standard items and then makes possible adjustments, to achieve an opposite approach that starts with tailor-made solutions for the individual application.

The second front of innovation concerns the production of pharmaceuticals and the overlapping of production and distribution phases: 3D printing can represent a solution for the production of medicines directly in the pharmacy, also taking into account the fact that the composition of drugs tends to be tailored to the individual patient’s pathology.

Therefore, it emerges that the adoption of 3D printers within the medical sector can lead to a change in the shape of both production and distribution processes. It has been previously discussed how the additive manufacturing technique has made changes in the development of products, especially at the production of prototypes level [38]. Current trends, as this study highlights, are showing a further stage of development of 3D printing, involving several phased of production and questioning the business models of the medical and pharmaceutical sectors so far established. Among the main consequences, there is the fact that the layer-to-layer production logic allows simplifying procedures and ensures reduced times [16,19].

As for the impact at the supply chain level, starting from the assumption that the biggest change introduced by the 3D printer concerns the displacement of production in space (allowing the creation of an object anywhere), this means that the number of intermediaries in going to be significantly reduced, favoring faster production and, consequently, a reduced time to market. Furthermore, benefits can be seen also in the reduction of stocks in warehouses, as well as a reduction of waste related to overproduction of materials.

With regard to the attributes related to current and future opportunities and barriers generated by the adoption of the aforementioned technologies in the medical sector (RQ2), the research shows consensus in relation to the development of highly personalized materials aimed at improving the lives of individuals and the possibility of new market outlets. Undoubtably, the development of 3D printers can bring to a series of improvements in medical applications, contributing to preserve people’s health and save lives. The new frontiers concern the possibility of 3D printing human organs such as heart, liver and lungs and, through the innovative bioprinting technique, the possibility of producing ad hoc human organs and tissues using human cells, reducing the risk of rejection and obviating the problem of long waiting lists in transplant centers. This aspect leads back to another opportunity positively valued by companies, related to the development of biocompatible materials. It still remains a challenge to ensure that bioprinted tissues properly match the structure and properties of native tissues.

Among the opportunities that 3D printing in the healthcare sector is going though, the relevant impact of the technology during the current pandemic period cannot be overlooked. 3D printers had a relevant role, thanks to networks of makers who have organized themselves as new manufacturers, and to universities which exploited facilities and staff’s skills. There has been a great effort and collaboration to address the pandemic and the acute need for PPE and medical equipment that could not be fulfilled by existing manufacturing capacity, due to increased lead times, lack of government planning and procurement of these equipment.

As regards the criticalities and weaknesses related to the 3D printing use, they appear to be perceived by companies as very weak and not very significant. It has been observed that the most important attributes concern the adequate training of personnel and the unregulated market in which they operate.

As regards the non-regulation of the market, the main international standardization bodies including the International Organization for Standardization (ISO) and the American Society for Testing Materials (ASTM) are working assiduously for the creation of technical standards that can regulate the sector. Similar efforts are expected from the European Union and individual states to regulate these activities by means of directives and laws.

This study has some limitations. First, is the fact that the perception of the impact of 3D printers has been analyzed on those who produce 3D printers. In future studies would be important to compare this perspective with that of users in the medical field to verify similarities and differences. Another limitation derives from the qualitative nature of the analysis, which allowed to carry out an in-depth comprehension of the impacts at the organizational and supply chain level, but on a limited base of respondents, who are located exclusively in Italy, being results not generalizable. Future research could firstly develop a quantitative study to help measuring and generalize these results, by considering a wider sample of companies, not only in Italy, in order to evaluate the situation in other European or worldwide countries. Another area for further research is to compare the use of traditional technologies with the use of additive manufacturing ones in the medical field, in order to evaluate advantages and disadvantages of the latter.

## Figures and Tables

**Table 1 ijerph-19-07057-t001:** Socio-demographic profile of companies.

	Business 1	Business 2	Business 3	Business 4	Business 5	Business 6	Business 7
**Business dimension**	10–49 employees	>250 employees	50–249 employees	1–9 employees	10–49 employees	10–49 employees	1–9 employees
**Income**	1–10 mln	11–50 mln	>50 mln	<1 mln	1–10 mln	11–50 mln	<1 mln
**Business** **location**	North Italy	Center Italy	North Italy	South Italy	North Italy	North Italy	North Italy
**Region**	Piedmont	Lazio	Veneto	Campania	Emilia-Romagna	Piedmont	Emilia- Romagna
**Reference markets**	Italy, Europe, International markets	Italy, Europe	Italy, Europe, International markets	Italy, Europe	Italy, Europe, International markets	Italy	Italy
**Geographical area of the international market**	North America	-	North America, Asia	-	North America	-	-
**Foundation year**	1955	2010	1961	2014	2012	1989	2017
**Born for 3D** **printing**	No, reconverted	No, reconverted	No, reconverted	No, reconverted	Yes	No, reconverted	No, reconverted

**Table 2 ijerph-19-07057-t002:** 3D printing technologies implemented.

	Business 1	Business 2	Business 3	Business 4	Business 5	Business 6	Business 7
Fusion Deposition Modelling (FDM) [35,36]	X	X		X	X		X
Selective laser sintering (SLS) [35,36]						X	
Stereolithography (SLA) [35,36]				X			
PolyJet 3D printing [35,36]							
3D printing with resin [35,36]			X	X			
Laminated object manufacturing (LOM) [36]							
Selective Laser Melting (SLM) [36]			X				
Fluid mix printing [36]					X		

The symbol X stand for a check mark to indicate that Business 1 has FDM as a 3D printing technique.

**Table 3 ijerph-19-07057-t003:** Businesses’ main customers.

	Business 1	Business 2	Business 3	Business 4	Business 5	Business 6	Business 7
University graduate schools (Faculty of medicine)		X		X	X		
Healthcare companies	X				X	X	X
Training centers							
Manufacturing businesses		X					
Dental prosthesis manufacturers			X				
Privates				X			

The symbol X stand for a check mark to indicate that Business 1 has FDM as a 3D printing technique.

**Table 4 ijerph-19-07057-t004:** Companies perception on the impact of the use of 3D printers in the medical sector at an organizational level.

	Business 1	Business 2	Business 3	Business 4	Business 5	Business 6	Business 7
Changes in tasks/ functions	4	5	3	3	3	3	5
Introduction of new roles and new tasks	4	3	3	4	5	2	5
Changes in procedures, routines and activities	4	3	3	4	4	5	5
Orientation to a new organizational culture	4	4	3	5	4	4	5

**Table 5 ijerph-19-07057-t005:** Companies’ perception on the impact of the use of 3D printers in the medical sector at a supply chain level.

	Business 1	Business 2	Business 3	Business 4	Business 5	Business 6	Business 7
Reduction of inventories	5	3	3	2	4	4	4
Internal logistics optimization	5	2	2	2	3	4	4
Time to market reduction	5	3	4	4	4	5	4
Reduction of actors in the supply chain	4	4	4	3	5	5	4

**Table 6 ijerph-19-07057-t006:** Current and future opportunities generated by the application of 3D printers in the medical field.

	Business 1	Business 2	Business 3	Business 4	Business 5	Business 6	Business 7
Development of highly customized materials aimed at improving people’s lives	5	5	4	5	3	5	5
Development of new and increasingly sophisticated technologies	5	4	4	5	4	3	3
Birth of new specialized professional figures	4	3	3	5	5	4	5
Development of biocompatible materials	5	4	4	5	3	4	5
Possibility of new market outlets	5	4	3	5	5	5	4

**Table 7 ijerph-19-07057-t007:** Companies perception on the negative impacts and criticalities generated by the use of 3D printers in the medical sector.

	Business 1	Business 2	Business 3	Business 4	Business 5	Business 6	Business 7
Strong investments for the purchase of machinery and equipment	3	3	2	2	2	2	3
Lack of maturity and reliability of the technology	3	4	2	3	3	2	2
Staff training	2	5	3	3	5	2	3
Unregulated market	5	5	3	2	5	2	3
Ethical and social issues	1	4	2	2	2	1	2

## Data Availability

The data are made available by the authors upon request.

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
