# Peer review of "Organizational and Supply Chain Impacts of 3D Printers Implementation in the Medical Sector"

_ijerph, 2022, doi:10.3390/ijerph19127057_

Round 1
Reviewer 1 Report
The additive manufacturing, also know as the 3D printing, is becoming an inseparable component in the manufacturing industries ranging from rapid prototyping of models to production of aerospace engines. To this end, the medical sector is hugely benefitted by 3D printed devices and accessories as it offers highly customised patient-specific solutions that are not simply possible with other manufacturing options.
In this manuscript, authors conducted a quantitative survey to understand the impact of 3D printed medical devices on the organisation and supply chain mechanisms. For this, after performing an introductory discussion, authors conducted interviews of seven different companies, mainly based in Italy. The organisation of the article is clear, well-written and very timely. Hence, I must recommend its acceptance after MINOR revisions. In this case, following points must be addressed:
- In Page 1, Line 42; It should be 'additive manufacturing, better known as 3D printing". Because additive manufacturing is the main term, 3D printing is part of it.
- Authors nicely mentioned the importance of 3D printed medical devices in the case of emergency needs. In this case, they must mention the successful use of the 3D printing facilities in the current Covid-19 pandemic. In fact, it started from Italy ! Refer 2-3 related references here in the introduction, https://www.sciencedirect.com/science/article/pii/S0278612520302351 and https://www.tandfonline.com/doi/abs/10.1080/16258312.2021.1932568
- Authors mentioned various 3D printing processes that are widely used in printing medical devices in Table 2. There must be references that define these 3D printing processes, e.g.
https://www.sciencedirect.com/science/article/pii/S2352940721003693
and https://pubs.acs.org/doi/10.1021/acs.chemrev.7b00074
Once authors take my suggestions into account, I will be happy to accept this excellent piece of work.
Author Response
Notes on the revision of the manuscript ID ijerph-1715088
Dear Editor, dear Reviewers
Thank you for giving us the opportunity to revise our paper and resubmit it to International Journal of Environmental Research and Public Health. Adhering to Reviewers’ recommendations, we have revised our manuscript by considering each comment.
We hope that you will find the enclosed version of the paper improved and suitable for publication in your Journal. Thank you very much for your efforts regarding our submission.
The Authors
REVIEWER 1
Comment 1:
The additive manufacturing, also known as the 3D printing, is becoming an inseparable component in the manufacturing industries ranging from rapid prototyping of models to production of aerospace engines. To this end, the medical sector is hugely benefitted by 3D printed devices and accessories as it offers highly customised patient-specific solutions that are not simply possible with other manufacturing options.
In this manuscript, authors conducted a quantitative survey to understand the impact of 3D printed medical devices on the organisation and supply chain mechanisms. For this, after performing an introductory discussion, authors conducted interviews of seven different companies, mainly based in Italy. The organisation of the article is clear, well-written and very timely. Hence, I must recommend its acceptance after MINOR revisions. In this case, following points must be addressed:
Response 1:
Thank you very much for appreciating our work, we have tried to do our best to follow Reviewers’ suggestions, and we hope that you will find the new version of the manuscript improved and valuable of publication.
Comment 2:
In Page 1, Line 42; It should be 'additive manufacturing, better known as 3D printing". Because additive manufacturing is the main term, 3D printing is part of it.
Response 2:
Thank you very much for this suggestion, we agree with your comment and we have changed it as requested.
Comment 3:
Authors nicely mentioned the importance of 3D printed medical devices in the case of emergency needs. In this case, they must mention the successful use of the 3D printing facilities in the current Covid-19 pandemic. In fact, it started from Italy ! Refer 2-3 related references here in the introduction, https://www.sciencedirect.com/science/article/pii/S0278612520302351 and https://www.tandfonline.com/doi/abs/10.1080/16258312.2021.1932568
Response 3:
Thank you for your suggestion, we agree that it is relevant to cite the use of 3D printing also in the case of Covid-19 pandemic; we have added it considering also the references suggested in the introduction section, and as a comment in the conclusion one.
Comment 4:
Authors mentioned various 3D printing processes that are widely used in printing medical devices in Table 2. There must be references that define these 3D printing processes, e.g. https://www.sciencedirect.com/science/article/pii/S2352940721003693 and https://pubs.acs.org/doi/10.1021/acs.chemrev.7b00074
Response 4:
Thank you very much for your comment. We have referenced all the 3D printing processes present in Table 2 as suggested by the Reviewer.
Reviewer 2 Report
The topic and the two questions mentioned in introduction are meaningful. However, the authors do not offer some valuable ideas in the conclusion section, but something that is generally known. In the conclusion section, explaining some common definitions such as culture, and citing others' viewpoints are not the way to write this section.
In terms of experimental methods, the important changes in the introduction of 3D printing into the medical field should be compared with the traditional technology, rather than the self-comparison within 3D printing technologies. The experimental data lacks quantitative analysis and cannot give the author an intuitive feeling.
Author Response
Notes on the revision of the manuscript ID ijerph-1715088
Dear Editor, dear Reviewers
Thank you for giving us the opportunity to revise our paper and resubmit it to International Journal of Environmental Research and Public Health. Adhering to Reviewers’ recommendations, we have revised our manuscript by considering each comment.
We hope that you will find the enclosed version of the paper improved and suitable for publication in your Journal. Thank you very much for your efforts regarding our submission.
The Authors
REVIEWER 2
Comment 1:
The topic and the two questions mentioned in introduction are meaningful. However, the authors do not offer some valuable ideas in the conclusion section, but something that is generally known. In the conclusion section, explaining some common definitions such as culture, and citing others' viewpoints are not the way to write this section.
Response 1:
Thank you for this comment, we have tried to follow Reviewer’s advice by introducing in the conclusion section some more focused reflections, in order to provide a clearer contribution on the impact of 3D printing in the medical sector.
Comment 2:
In terms of experimental methods, the important changes in the introduction of 3D printing into the medical field should be compared with the traditional technology, rather than the self-comparison within 3D printing technologies. The experimental data lacks quantitative analysis and cannot give the author an intuitive feeling.
Response 2:
Thank you for your comment. In this study we have decided to focus only on 3D printing technologies without comparing it with the traditional ones. It could be relevant to take your suggestion for further studies and compare traditional technologies with AM. We have inserted this as a main future research direction. As for the data of our analysis, we know that the nature of the methodology used (qualitative with 7 case studies) cannot help to generalize the results obtained, even if it permits to have a widen qualitative perspective of AM in the sector. We have underlined this as a main limitation of the study, with relevance for further research to widen the sample considered in order to have a confirmation of the results obtained.
Reviewer 3 Report
The aim of this study is to understand the way in which the adoption of medical 3D printers has caused economic-business changes at the supply chain, organizational and environmental level within business processes. Using semi-structured interviews administered to seven Italian companies that design, manufacture, and sell 3D printers offering additive technologies to the medical sector, a multiple case study was developed. However, there are several issues that need to be addressed.
-
In Table 3, it shows that Business 2,3,4 do not have Healthcare companies as their main customers. In such a case, how can you verify that the data collected is accurate?
-
What is the size of the sample? How many samples will be collected from each company? Please explain the sample size in detail.
-
The method and analysis used in this study are not convincing to validate the results unless further evidence is presented.
Author Response
Notes on the revision of the manuscript ID ijerph-1715088
Dear Editor, dear Reviewers
Thank you for giving us the opportunity to revise our paper and resubmit it to International Journal of Environmental Research and Public Health. Adhering to Reviewers’ recommendations, we have revised our manuscript by considering each comment.
We hope that you will find the enclosed version of the paper improved and suitable for publication in your Journal. Thank you very much for your efforts regarding our submission.
The Authors
REVIEWER 3
Comment 1:
The aim of this study is to understand the way in which the adoption of medical 3D printers has caused economic-business changes at the supply chain, organizational and environmental level within business processes. Using semi-structured interviews administered to seven Italian companies that design, manufacture, and sell 3D printers offering additive technologies to the medical sector, a multiple case study was developed. However, there are several issues that need to be addressed.
In Table 3, it shows that Business 2,3,4 do not have Healthcare companies as their main customers. In such a case, how can you verify that the data collected is accurate?
Response 1:
It is true that they not have health care companies as main customers, but this not means that their customers do not work in the healthcare sector, in fact, Business 2 and 4 works with universities of medicine, therefore they provide 3D printing for their healthcare laboratories, while Business 3 works with Dental prosthesis manufacturers, that are a branch of the healthcare system. In Table 3 we have detailed that when speaking about universities, we refer to “Faculties of medicine”.
Comment 2:
What is the size of the sample? How many samples will be collected from each company? Please explain the sample size in detail.
Response 2:
As written in the methodology section the sample size is represented by 7 companies. The decision to analyze 7 companies in a qualitative analysis is in line with Yin (2003), according to which, the recommended number of units to be examined to analyze a good multiple case study is between 4 and 12. We have interviewed the responsible figure for 3D printing production for each company. As written in the paper “these were carried out with the Chief Executive Officers of the participating companies, or for smaller companies with the owner, looking for interviewing the position with the greatest technical and organizational skills of the company”.
Comment 3:
The method and analysis used in this study are not convincing to validate the results unless further evidence is presented.
Response 3:
Thank you for your comment. We know that the nature of the methodology used (qualitative with 7 case studies) cannot help to generalize the results obtained. However, it allows the emergence of hypotheses that may have not been previously considered, thus representing the basis for subsequent quantitative analyzes, aimed at verifying and measuring the hypotheses identified. Anyway, we have underlined this as a possible limitation of the study, with the indication that future studies should be addressed at confirming and measuring the results obtained.
Round 2
Reviewer 2 Report
The article has been improved.
Reviewer 3 Report
No further comments.